# Evaluation of Patients’ Levels of Walking Independence Using Inertial Sensors and Neural Networks in an Acute-Care Hospital

**DOI:** 10.3390/bioengineering11060544

**Published:** 2024-05-26

**Authors:** Tatsuya Sugimoto, Nobuhito Taniguchi, Ryoto Yoshikura, Hiroshi Kawaguchi, Shintaro Izumi

**Affiliations:** 1Department of Rehabilitation, Japanese Red Cross Kobe Hospital, Kobe 651-0073, Japan; 2Graduate School of System Informatics, Kobe University, Kobe 657-8501, Japan; 3Graduate School of Science Technology and Innovation, Kobe University, Kobe 657-8501, Japanshin@cs28.cs.kobe-u.ac.jp (S.I.); 4Osaka Heat Cool Inc., Osaka 562-0035, Japan

**Keywords:** neural network, inertial sensor, level of walking independence, 10-m walk test, timed up-and-go test

## Abstract

This study aimed to evaluate walking independence in acute-care hospital patients using neural networks based on acceleration and angular velocity from two walking tests. Forty patients underwent the 10-m walk test and the Timed Up-and-Go test at normal speed, with or without a cane. Physiotherapists divided the patients into two groups: 24 patients who were monitored or independent while walking with a cane or without aids in the ward, and 16 patients who were not. To classify these groups, the Transformer model analyzes the left gait cycle data from eight inertial sensors. The accuracy using all the sensor data was 0.836. When sensor data from the right ankle, right wrist, and left wrist were excluded, the accuracy decreased the most. When analyzing the data from these three sensors alone, the accuracy was 0.795. Further reducing the number of sensors to only the right ankle and wrist resulted in an accuracy of 0.736. This study demonstrates the potential of a neural network-based analysis of inertial sensor data for clinically assessing a patient’s level of walking independence.

## 1. Introduction

The recent global aging trend has led to an increased number of elderly patients admitted to acute-care hospitals. Consequently, many patients exhibit inherent walking impairments due to underlying illnesses and frailty [1]. Given the risk of further decline in walking function resulting from post-hospitalization treatments and prolonged bed rest, it becomes imperative to maintain or restore walking function through appropriate rehabilitation assessments and exercise therapies [2,3]. One approach to increasing ambulation practice during hospitalization involves physiotherapists allowing patients in the process of regaining ambulation to walk within the ward, either under the supervision of medical staff or independently. Thus, promoting ambulation independence at an appropriate juncture is significant for mitigating the adverse effects of rest-related disuse syndrome and facilitating early hospital discharge. However, switching to walking with a cane or without an aid can be complex for patients with advanced walking with a walker. This is because, unlike using a walker, walking with a cane or no aid increases the risk of falls owing to the reduced support surface.

The 10-m walk test (10 MWT) and Timed Up-and-Go test (TUG) are commonly used in clinical practice to assess gait function objectively. The 10 MWT involves walking 10 m in a straight line at a comfortable pace, with the time taken to calculate walking speed using a stopwatch. Previous research on community-dwelling elderly individuals has identified a 10 MWT time of 10 s or longer, corresponding to a walking speed cutoff value of 1.0 m/s or less, as a diagnostic criterion for frailty [4,5] and a predictor of long-term care needs [6]. Conversely, the TUG starts from a seated position and involves rising from the chair upon cue, walking at a comfortable or maximum speed to a designated point 3 m away, changing direction at that point, returning to the chair, and measuring the time taken to complete the task [7]. Research on community-dwelling older adults has associated a TUG cutoff value of 13.5 s or more with an increased risk of falls in daily living [8]. Therefore, shorter completion times for both tests indicated better walking and balance functions. However, it is essential to note that the hospital environment, a controlled space without steps or obstacles, may not necessarily require patients to meet these cutoff values to determine their level of walking independence. In clinical practice, physiotherapists may permit patients with slower movement speeds who do not meet the cutoff values to walk in the hospital ward after evaluating their movement techniques. Thus, the assessment of the level of walking independence considers not only movement speed but also the actual method of movement, highlighting the importance of subjective judgment by physiotherapists. However, the evaluation of the movement techniques lacks objectivity.

Various devices are available to assess movements objectively; however, inertial sensors are simple and versatile. In this approach, a small and lightweight sensor is affixed to the patient’s body, and the acceleration and angular velocity are recorded during walking to evaluate movement characteristics. Numerous prior studies have utilized inertial sensors to evaluate the 10 MWT and compare healthy individuals with patients [9,10,11], but few studies have compared differences in physical function among patients. For instance, one study examined patients with cerebrovascular disease (CVD) within 1–6 months after illness onset [12]. In this study, patients wore inertial sensors on the head, chest, and lower back and were categorized into monitoring and independent groups based on their level of walking independence. The results indicated that patients in the monitoring group exhibited significantly lower acceleration amplitudes and symmetry in these three regions than those in the independent group. Similarly, inertial sensors have been employed in the TUG, enabling data segmentation from the lumbar and ankle regions into six sub-phases: sit-to-stand, two walking phases, two turning phases, and stand-to-sit [13]. A study comparing different fall risks in patients with CVD at least 6 months post-onset found that those requiring more than 20 s to complete the TUG exhibited significantly longer walking and turning times as well as lower angular velocities during turning [14].

Therefore, the acceleration and angular velocity data during these tests reflect the clinical fall risk and level of walking independence. Hence, inpatients with various diseases undergoing rehabilitation interventions performed the TUG at maximum speed using an inertial sensor attached to the lower back [15]. Patients were categorized into three groups based on their level of walking independence and a TUG time cutoff of 13.5 s. Results revealed that, consistent with previous findings, the monitoring and independent groups requiring more than 13.5 s demonstrated significantly longer total TUG times than those requiring less than 13.5 s. However, differences in walking time and angular velocity during turning were only observed between the monitoring group and the independent group, which required less than 13.5 s, with no significant differences observed between the latter and the independent group, which required more than 13.5 s. This suggests that, even among patients with slower movement speeds, distinctions based on their level of walking independence were discernible.

In recent years, acceleration and angular velocity data analysis using machine learning, especially neural networks (NN), have been increasingly used for further data analysis. This approach allows for a more comprehensive and detailed evaluation of gait function, including the interrelationships among different measurement items. In a previous study involving healthy subjects, smartwatches with built-in inertial sensors were worn on the left and right wrists [16]. The participants were instructed to perform a 15-m walk at normal speed with and without visual field restrictions. The classification accuracies of these conditions were compared using four methods: random forest (RF), support vector machine (SVM), convolutional neural network (CNN), and recurrent neural network (RNN). The results indicated that the CNN and RNN models outperformed traditional machine learning methods. Another study investigated the use of NN to predict differences in fall risk among community-dwelling elderly individuals aged 65 years or older [17]. The participants performed the TUG with inertial sensors attached to their necks and feet. The data were analyzed using SVMs and CNN, with the CNN model showing higher sensitivity, particularly when analyzing the angular velocity of the neck. Similarly, in another study, a 55-year-old community resident performed the TUG with an inertial sensor attached to the waist [18]. The fall risk level classification accuracy was compared among machine learning methods such as RF, SVM, and CNN. The CNN model exhibited the highest accuracy.

These findings suggest that NN-based analysis can be applied to classify walking independence among hospitalized patients, as the acceleration and angular velocity during walking tasks vary depending on the patient’s physical function and fall risk. However, no such studies have been conducted to date. Therefore, this study aimed to develop an NN model to evaluate the level of walking independence of patients admitted to an acute-care hospital based on acceleration and angular velocity data collected during the 10 MWT and TUG. The anticipated outcome was that the model’s accuracy would be comparable to that of judgments made by physiotherapists.

## 2. Materials and Methods

### 2.1. Patients

A total of 40 patients (mean age 78.2 years, range 52–94 years) were enrolled in this study, comprising 19 men and 21 women. These patients were admitted to the acute-care hospital for emergency or surgical purposes related to medical and surgical conditions and subsequently received rehabilitation intervention. The inclusion criteria were as follows: (1) patients who were independently ambulating with a cane or unaided before admission; (2) patients who were capable of ambulating with a walker, at least independently, at the time of measurement; and (3) patients who had initiated practicing walking with a cane or no aid within one week. The exclusion criteria included evident cognitive decline, comorbidities, or medical history that would impede gait practice or measurement.

Patients were classified into two groups: the “Walking Acquired” (WA group) comprised those who ambulated with a cane or unaided within the ward, with or without monitoring, while the “Not Acquired” (NA group) included patients who still required a walker for ambulation.

### 2.2. Measurement Procedure

Before testing, using Velcro bands, eight inertial sensors (Xsens DOT, Xsens) equipped with built-in triaxial acceleration and angular velocity sensors were attached to the patient’s body. The sensors were affixed to the chest over the sternum lower back at the level of the third lumbar vertebra, above the right and left wrists, midfront of the right and left thighs, and above the right and left ankles. These sensors had a measurement range of 16 G for acceleration and 2000 °/s for angular velocity, with a sampling frequency of 120 Hz. Six acceleration and angular velocity signals were continuously recorded for each sensor during both 10 MWT and TUG. The signals include three acceleration axes: anterior-posterior acceleration (X-axis), mediolateral acceleration (Y-axis), and vertical acceleration (Z-axis), and three angular velocity axes: roll, which is the rotation around the X-axis; pitch, which is the rotation around the Y-axis; and yaw, which is the rotation around the Z-axis. The sensors were connected to a tablet device (iPad Mini, Apple) via Bluetooth and operated using a dedicated application.

After explaining the method of each test to the subjects, we commenced the 10 MWT. A straight line of 16 m, including acceleration and deceleration paths of 3 m each, was used, and the time required to cover the central 10 m was measured using a stopwatch. Subsequently, the TUG was performed. The TUG used a chair with a seat height of 43 cm and one cone. Using a stopwatch, we measured the time required to stand up from the chair, walk 3 m, turn at the cone, return, and sit down again. Figure 1 shows an actual TUG measurement scene with an attached inertial sensor. Both tests were conducted twice at normal speed, and the patients were instructed to walk with or without a cane. The decision regarding the mode of walking was made in consultation with the physiotherapists in charge and the patients themselves. Other physical functions assessed included grip strength and the Frailty Screening Index (FSI) [19]. Patients were also asked about any falls they may have experienced in the past year.

### 2.3. Signal Processing

After the measurement, the data from each sensor were transferred to a laptop using dedicated software to synchronize the data between the sensors. The 10 MWT data from both trials were analyzed, excluding the first and last steps and including data from the acceleration and deceleration paths. The TUG followed the method of a previous study [13], with the sub-phases of sit-to-stand and stand-to-sit, as well as the start and end points of the two turning phases, identified. Specifically, the sit-to-stand and stand-to-sit phases were identified by the maximum and minimum values of the pitch angle of the lumbar angular velocity, respectively (see the pitch angle in the lower part of Figure 2). Additionally, the two turning phases were identified as the maximum absolute value of the yaw angle, followed by locating the range before or after this value and below a threshold value of 0.1 (see the yaw angle in the lower part of Figure 2). The first walking phase was defined as the period from the end of the sit-to-stand phase to the start of the first turn. The second walking phase was defined as the period from the end of the first turn to the start of the second turn. In this study, only these walking phases were used in both tests to analyze the data while walking in conjunction with the 10 MWT.

Pre-processing for use with the NN input data was performed as follows: First, the acceleration data were high-pass filtered at 1 Hz to exclude the effects of gravitational acceleration. Next, to use the data for each gait cycle, the combined accelerations of the right and left ankles, as shown in Equation (1), were calculated, respectively. These calculations were then differentiated from one sample neighbor, and the periodic peak value was defined as the initial contact (IC) for each cycle (Figure 3). For example, if the right gait cycle is 4 steps in the first walking phase of the TUG and three steps in the second phase, seven data sets for the right gait cycle were obtained. This study used left gait cycle data in the analysis because more left gait cycle datasets were obtained than right.
(1)$$Combined\;Acceleration=\sqrt{Acceleration_{X-axis}^{2}+Acceleration_{Y-axis}^{2}+Acceleration_{Z-axis}^{2}}$$

Furthermore, as a method of normalization, the acceleration and angular velocity were set to a range of 0 to 1 so that the baseline of each dataset was 0.5, as they varied in both the positive and negative directions. Specifically, for each of the three axes of acceleration and angular velocity for each subject, the value of each sensor was divided by the maximum absolute value of all sensor values, then by 2, and a 0.5 was added. Depending on whether the maximum value used for the division is positive or negative, the normalized data will satisfy either a minimum value of zero or a maximum value of one for any sensor in any gait cycle within a trial. Finally, to align the lengths of all data sets, a 0.5 was added to the end of the data set shorter than 276, the longest walking cycle (Figure 4). All the data processing steps were performed using Python and its libraries.

### 2.4. Neural Network

Transformer was implemented using PyTorch version 2.0.0, a Python library. The Transformer is a machine-learning model [20] known for its strong performance in natural language processing and image recognition tasks. It is built around an attention mechanism (attention) that selectively highlights important feature information by computing the association between each input element and the other elements. This enabled the model to effectively capture long-distance dependencies and flexibly learn feature combinations at different locations. The architecture of the Transformer model used in this study included an input layer, Input Embedding, Positional Encoding, Transformer Block, Fully Connected (FC) layer, Softmax Function, and an output layer. The Transformer Block comprises a multihead attention module, feed-forward module, Layer Normalization, and residual connection, which were repeated five times in this study for deeper learning. The program defined a class using nn.Module and nn.TransformerEncoder, which incorporated the embedding layer, Positional Encoding layer, Transformer’s encoder layer, and FC layer. The structure of the Transformer network is illustrated in Figure 5, and the specifics of the parameters for each network layer are listed in Table 1. The input and output sizes from the input data to the Transformer Block in Table 1 represent the batch size, data length, and number of channels, respectively. Details regarding the batch size and number of channels are described below.

To augment the total input data, this study included the original dataset and four additional datasets, each with random noise ranging from −0.1 to 0.1 added to every value. This approach resulted in a five-fold increase in data points compared with the original dataset. The data ratio between the two groups was balanced in the training and test datasets, with a test data ratio of 0.1. The training process utilized 130 epochs, a batch size of 64, and a learning rate of 1 × 10^−5^. The final accuracies of the training and test datasets were recorded. For evaluation, the leave-one-out-of-one cross-validation (LOOCV) method was employed, wherein each subject was iteratively excluded from the training, and the resulting model was used to predict the omitted subject. This process yielded 40 models, corresponding to the number of subjects in the dataset. A confusion matrix was generated for each inference according to Table 2, and the accuracy, sensitivity, and specificity, as expressed in Equations (2)–(4), were calculated. In the present study, the NA group was defined as positive.
(2)$$Accuracy=\frac{TP+TN}{TP+TN+FP+FN}$$
(3)$$Sensitivity=\frac{TP}{TP+FN}$$
(4)$$Specificity=\frac{TN}{TN+FP}$$

The input data consisted of 48 channels comprising six axes of acceleration and angular rate meters for each of the eight sensors per subject. To propose a more straightforward method for clinical applications, this study assessed the model’s performance with a reduced number of sensors. Specifically, the average accuracy of LOOCV was evaluated by excluding one sensor at a time. When excluded, five sensors were identified as important sites based on a significant decline in accuracy. Subsequently, two to five sensors were used for validation. All excluded sensor data were replaced with a value of 0.5 to maintain consistency.

### 2.5. Statistical Analysis

Statistical methods were used to compare basic information between the groups. The chi-square or Fisher’s exact test was used to compare categorical variables such as sex ratio, outcome (discharge or transfer), and history of falls in the past year. For other items, an unpaired *t*-test or Mann-Whitney U-test was used after checking each data point’s normal distribution and homogeneity of variance for group comparison. Statistical analyses were performed using a two-tailed test with Modified R Commander version 4.2.2 [21]. The level of statistical significance was set at a *p* value < 0.05. Descriptive statistics (means and standard deviations) were used to summarize the results.

## 3. Results

All trials were completed without adverse events in 26 and 14 patients in the WA and NA groups, respectively. The patient characteristics for each group are shown in Table 3. The diseases included spinal compression fractures, postoperative femoral neck fractures, postoperative knee or hip osteoarthritis (OA), postoperative lumbar spinal canal stenosis (LCS), CVD, and other medical conditions. The WA group was significantly younger than the NA group, had a significantly higher proportion of patients discharged home as the outcome destination, and the 10 MWT and TUG times were significantly shorter. The walking speeds calculated from the 10 MWT times of the WA and NA groups averaged 0.85 m/s and 0.61 m/s, respectively. There were no differences in the number of days from admission to measurement or the length of hospital stay between the groups.

Because two patients had missing TUG measurement data, we analyzed 40 patients for the 10 MWT and 38 patients for the TUG. The total number of gait cycles analyzed was 1684, with 981 in the WA and 703 in the NA groups. The dataset size was increased fivefold at each training iteration, resulting in approximately 8000 datasets. When trained with all eight sensors, the accuracy, sensitivity, and specificity were 0.836, 0.876, and 0.780, respectively. The confusion matrix and inference results for each subject are shown in Figure 6. Most subjects had an accuracy of 0.5 or better, but subjects 18 in the WA group and 1 and 3 in the NA group did not improve with further changes in the hyperparameters during learning. Next, we checked accuracy by excluding one sensor at a time to identify the critical sensor attachment sites for learning and inference. The results show that the accuracy decreased significantly for the right ankle (0.709), left wrist (0.730), right wrist (0.735), right thigh (0.801), and chest (0.812), as shown in Figure 7. Training with these five sensors yielded a slightly lower accuracy of 0.830, whereas the specificity improved to 0.821 compared to all eight sensors (Figure 8). Using three sensors (right ankle, left wrist, and right wrist) resulted in an accuracy of 0.795, which was lower than that achieved using five sensors (Figure 9). Finally, when comparing the accuracies of the two sensor combinations (right ankle/left wrist and right ankle/right wrist), the latter was superior, with an accuracy of 0.736 (Figure 10). The four LOOCV performance metrics are listed in Table 4.

## 4. Discussion

In this study, 40 patients admitted to an acute-care hospital in the early stages of gait practice after beginning cane or unassisted walking were fitted with eight inertial sensors to perform the 10 MWT and TUG. Acceleration and angular velocity while walking were measured to evaluate the patient’s level of walking independence using Transformer. The results showed that the accuracy was 0.836 using all sensors, although this was not as good as the evaluation by the physiotherapists. Even when the number of sensor sites was reduced to three and two, the accuracies remained at 0.795 and 0.736, respectively.

In a previous study analyzing inertial sensor data using an NN, 101 community-dwelling elderly people with an average age of approximately 75 years were assessed by clinicians for fall risk, achieving an accuracy of 0.865 based on the angular velocity of the neck during the TUG using a CNN [17]. Additionally, in predicting the occurrence of falls in 73 nursing home residents with an average age of approximately 83 years, a CNN achieved an accuracy of 0.750 using the acceleration and angular velocity of the lower back during a 6-min walk test [22]. Despite the small sample size of the present study (40 patients) and the comparable mean age, the achieved accuracy was similar to that of these studies. On the other hand, the physiotherapists’ assessment of the patient’s level of walking independence was based on an overall assessment of the time required for the two walking tasks and the actual method of movement. Nevertheless, there were significant group differences in age and time required for the 10 MWT and TUG. This reflects the physical functional decline due to aging and suggests that minimal walking speed may also be important for the possibility of independence.

Additional analyses were conducted for three subjects, 18 in the WA group and one and three in the NA group, who exhibited poor accuracy even with all sensors. Specifically, the first half of the walking phase of these three subjects (corresponding to the first half of the total number of steps of the 10 MWT and the first walking phase of the TUG) and data from all other subjects were used for training. In contrast, the second half of the walking phase of these three subjects (representing the second half of the total number of steps of the 10 MWT and the second walking phase of the TUG) was used to assess accuracy. Although the data extraction points differed, both training and inference involved information from the same subjects. Consequently, the accuracy of the number 18 in the WA group and number 1 in the NA group notably improved, whereas the accuracy of the number 3 in the NA group remained largely unchanged. This suggests that the poor accuracy of the two subjects may be due to the limited sample size. Conversely, it is possible that the motor skills of subject #3 in the NA group were sufficiently high to enable him to practice walking in the ward; however, this capability may not have been recognized. This oversight could be partly attributed to the limited time for physiotherapists to conduct repeated assessments. This study relied on data collected within the first week of walking practice, reflecting the participants’ level of walking independence at that specific time.

When the accuracy of LOOCV was examined by excluding one sensor at a time, a significant decrease in accuracy was observed, particularly when the sensors attached to the right foot, left hand, and right hand were excluded. This decrease can be attributed to the crucial role of the right lower limb in the left gait cycle. Specifically, in the left gait cycle, the right lower limb underwent an initial swing phase, followed by the IC and stance phases [23]. The range of motion was greater during the swing phase than during the stance phase because of the forward swing of the limb. The IC phase also involved the lower leg muscles, which supported the floor reaction force [23]. Hence, it is plausible that the acceleration and angular velocity of the contralateral lower limb, encompassing these phases and the transitional phase with a significant range of motion and floor reaction force, are pivotal in determining the level of walking independence. In contrast, the transition of the left lower limb from the stance to the swing phase appeared less critical. This finding is consistent with the fact that instability during gait tends to occur at the IC or shortly after that in clinical situations. Although the ankle is a common site analyzed in previous studies [10,11,15,17], including its relationship with the gait cycle is a novel finding.

Similar to the results for the ankle sensors, the accuracy decreased when sensors attached to the left and right wrists were excluded; however, this could be attributed to the fact that the wrists are also endpoints with a considerable range of motion. The arms swing during walking, as calculated from the inertial sensor data attached to the wrists, decreases with slower walking speed [24] and aging [25]. Additionally, individuals with a history of falls may not increase their arm swing in response to increased walking speed [26]. Therefore, wrist data can reflect a person’s gait function. However, most previous studies that analyzed the inertial sensor data using NN in elderly individuals and patients focused on sensors attached to the waist or feet. To our knowledge, only one study validated wrist sensor data conducted on healthy young adults [19]. This study achieved a classification accuracy of 0.889 for the two visibility conditions, although no sensors were worn at other locations. Therefore, the results of our study suggest that wrist acceleration and angular velocity may be essential factors in determining a patient’s level of walking independence.

In contrast, data from the lower back, which has been a focal point in prior studies [18,22], exhibited less significance in determining the level of gait independence in this study than data from the wrist and ankle. This discrepancy can be attributed to the normalization method employed in this study. Specifically, the maximum value among all eight normalization sensors consistently originated from the wrist or ankle across all subjects. Consequently, relatively minor acceleration and angular velocity fluctuations were observed in the chest and lower back, as illustrated in Figure 4. This normalization method was selected based on the premise that the interplay between acceleration and angular velocity across different anatomical sites plays a crucial role in determining walking independence. Therefore, modifying the normalization method, such as by adopting regional normalization, may yield diverse outcomes in future investigations.

The limitations of this study include an insufficient sample size and the fact that only walking cycles were used. Concerning the former, further validation with a larger sample size is needed before the results of this study can be generalized. In particular, the latter excluded the sit-to-stand, two-turn, and stand-to-sit sub-phases, which could have provided more critical information. However, focusing on the walking cycle allowed us to consider which body parts were crucial for the classification and why. As in previous studies, future analyses based solely on TUG data should encompass all phases.

## 5. Conclusions

Through the utilization of eight inertial sensors on hospitalized patients and the analysis of data from the 10 MWT and TUG walking phases with Transformer, one of the NN, we achieved notable accuracy in determining the patient’s level of walking independence, with an accuracy of 0.836. This accuracy closely approximated the evaluation accuracy achieved by physiotherapists. Furthermore, the accuracy was maintained at 0.795, even when the number of sensors was limited to three, positioned at both wrists and right ankle. This approach is valuable for mitigating the patient burden during measurements.

Based on the results of this study, this analysis method can support physiotherapists in clinical settings. Specifically, the inference results of the model can supplement the assessment findings of physiotherapists who struggle with deciding whether to allow patients to practice walking in the ward. Additionally, it can be beneficial when a different or less experienced physiotherapist than the one in charge evaluates a patient’s walking.

## Figures and Tables

**Figure 1 bioengineering-11-00544-f001:**
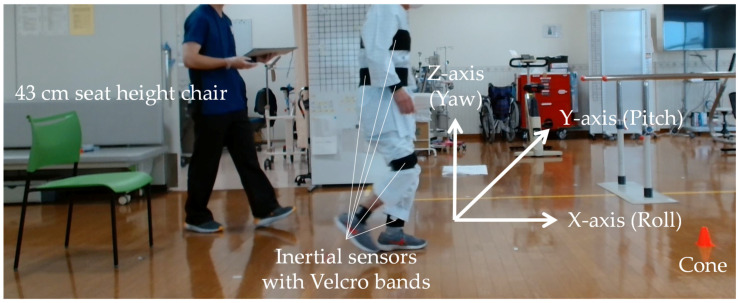
Sensor attachment sites and TUG measurement scene.

**Figure 2 bioengineering-11-00544-f002:**
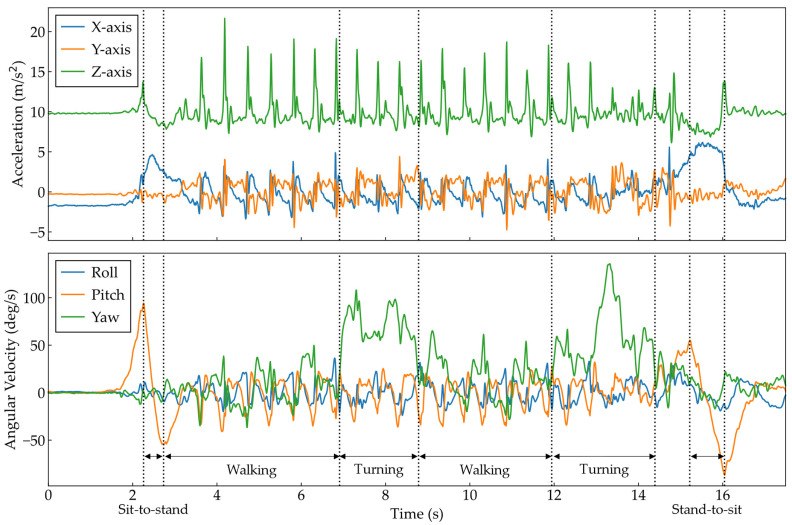
Example of acceleration and angular velocity of the lumbar during TUG.

**Figure 3 bioengineering-11-00544-f003:**
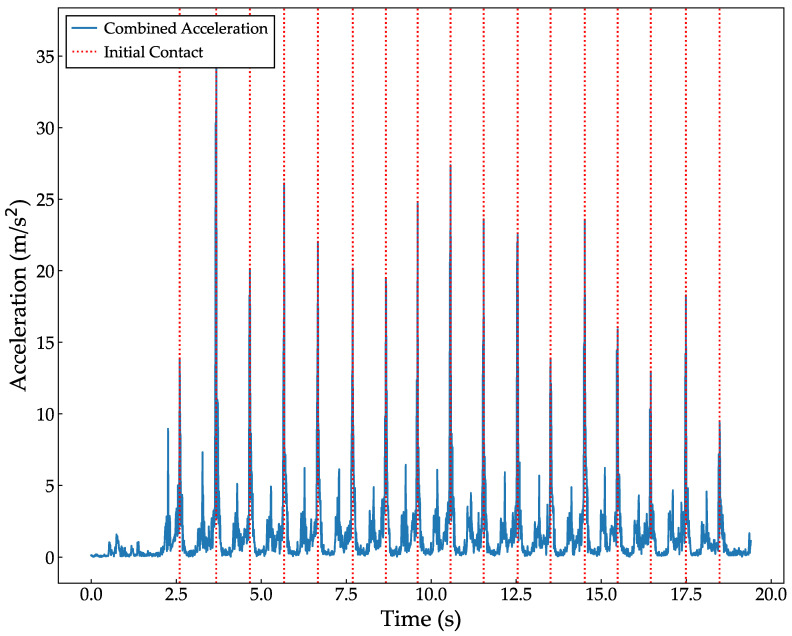
Example of identifying ICs from the combined acceleration of the ankle during 10 MWT.

**Figure 4 bioengineering-11-00544-f004:**
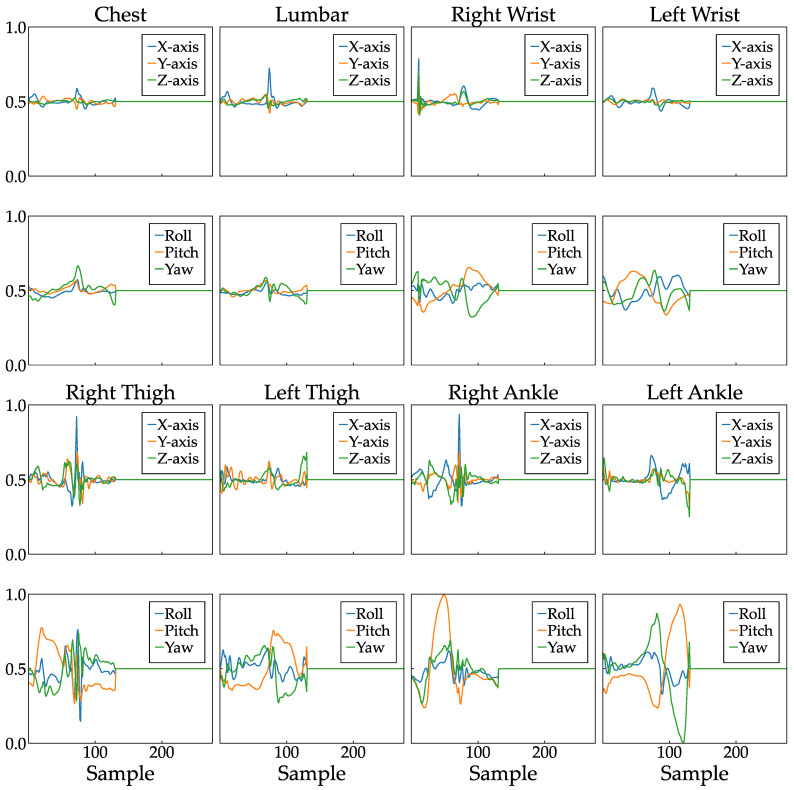
Example of normalized data for all 48 channels in one left gait cycle.

**Figure 5 bioengineering-11-00544-f005:**
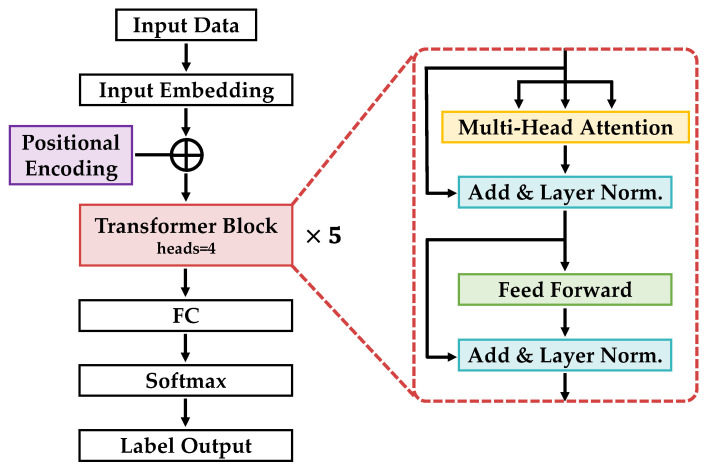
Transformer network architecture diagram.

**Figure 6 bioengineering-11-00544-f006:**
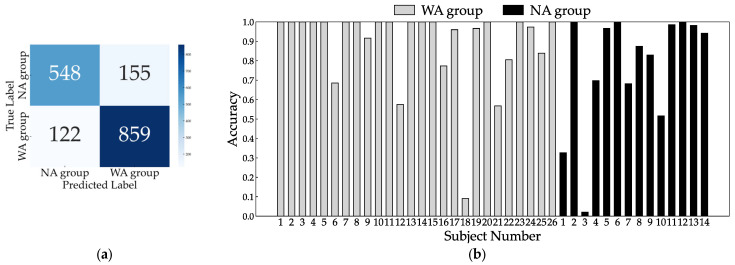
Confusion matrix (**a**) and individual inference results (**b**) of LOOCV using all sensors.

**Figure 7 bioengineering-11-00544-f007:**
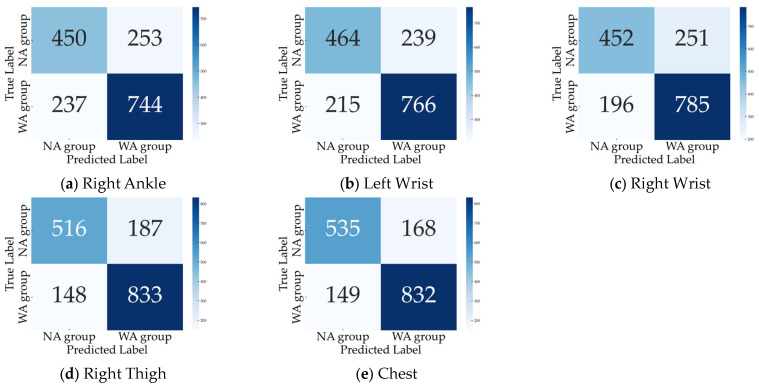
Confusion matrix by excluding (**a**) right ankle, (**b**) left wrist, (**c**) right wrist, (**d**) right thigh, and (**e**) chest.

**Figure 8 bioengineering-11-00544-f008:**
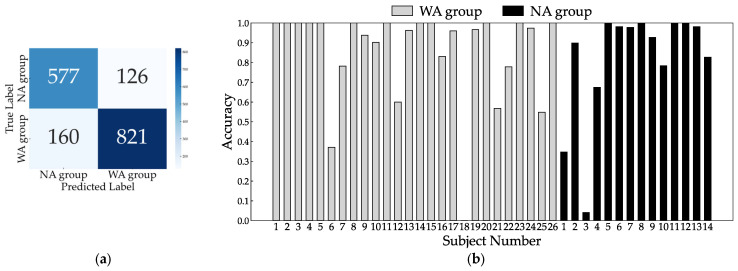
Confusion matrix (**a**) and individual inference results (**b**) of LOOCV using five sensors: right ankle, left wrist, right wrist, right thigh, and chest.

**Figure 9 bioengineering-11-00544-f009:**
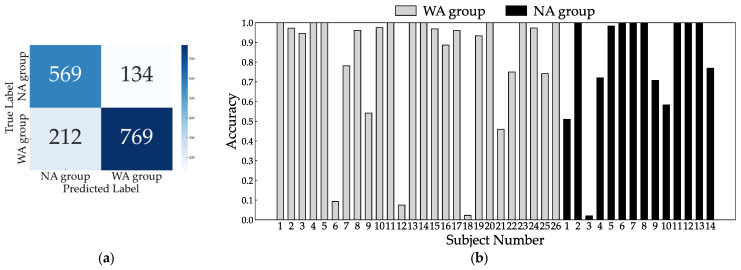
Confusion matrix (**a**) and individual inference results (**b**) of LOOCV using three sensors: right ankle, left wrist, and right wrist.

**Figure 10 bioengineering-11-00544-f010:**
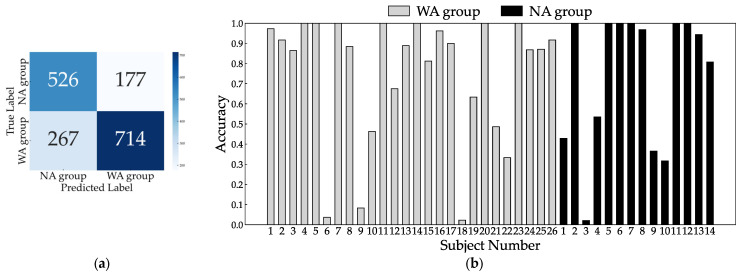
Confusion matrix (**a**) and individual inference results (**b**) of LOOCV using two sensors: right ankle and right wrist.

**Table 1 bioengineering-11-00544-t001:** Detailed parameters for each network layer.

Layer	Input	Output
Input Embedding	(64, 276, 48)	(64, 276, 48)
Positional Encoding	(64, 276, 48)	(64, 276, 48)
Transformer Block	(64, 276, 48)	(64, 276, 48)
FC	(64, 13248)	(64, 2)

**Table 2 bioengineering-11-00544-t002:** Confusion matrix.

**True Label**	**Predicted Label**	
	Positive (NA Group)	Negative (WA Group)
Positive (NA group)	True Positive (TP)	False Negative (FN)
Negative (WA group)	False Positive (FP)	True Negative (TN)

**Table 3 bioengineering-11-00544-t003:** Basic information for each group (M: male, F: female). Data are shown as the mean (standard deviation) except where noted. *p* values less than 0.05 are highlighted in bold.

	WA Group (*n* = 26)	NA Group (*n* = 14)	*p* Values
Age (years)	76.4 (10.3)	81.6 (3.8)	**0.029**
Gender	14 M, 12 F	5 M, 9 F	0.270
Height (cm)	158.1 (10.0)	156.5 (11.7)	0.671
Weight (kg)	56.5 (13.5)	58.4 (13.7)	0.672
Measurement since admission (days)	16.8 (8.5)	19.8 (7.1)	0.254
Total length of stay (days)	23.2 (8.1)	29.1 (10.2)	0.072
Discharge and transfer	22, 4	1, 13	**<0.001**
Diseases:			
Postoperative LCS	6	2	
Postoperative femoral neck fracture	2	6	
Postoperative knee or hip OA	5	1	
CVD	1	3	
Others	12	2	
10MWT time (s)	12.4 (2.8)	18.1 (6.4)	**0.006**
TUG time (s)	14.7 (2.9)	21.2 (6.6)	**0.003**
Grip strength (kg)	22.4 (8.8)	19.2 (10.4)	0.339
FSI (from 0 to 5)	1.5 (1.2)	2.1 (1.3)	0.148
Faller, %	5, 19.2	7, 50.0	0.071

**Table 4 bioengineering-11-00544-t004:** LOOCV performance metrics for each input data.

Input Data	Accuracy	Sensitivity	Specificity
All 8 sensors	0.836	0.876	0.780
5 sensors (right ankle, left wrist, right wrist, right thigh, and chest)	0.830	0.837	0.821
3 sensors (right ankle, left wrist, and right wrist)	0.795	0.784	0.809
2 sensors (right ankle and right wrist)	0.736	0.728	0.748

## Data Availability

The datasets presented in this article are not readily available because they are part of an ongoing study. Requests to access the datasets were directed to the first authors’ e-mail addresses.

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
