# Peer review of "Evaluation of Patients’ Levels of Walking Independence Using Inertial Sensors and Neural Networks in an Acute-Care Hospital"

_bioengineering, 2024, doi:10.3390/bioengineering11060544_

Round 1
Reviewer 1 Report
Comments and Suggestions for Authors
The paper is well written and of interest no major spelling or grammer errors detected
Author Response
I extend my sincere appreciation for your thorough evaluation of my manuscript. I am grateful for your time and commitment to this review process.
Reviewer 2 Report
Comments and Suggestions for Authors
This is an exciting paper. The paper evaluates walking independence in acute-care hospital patients using deep neural networks (based on a transformer model). The study used a large number of subjects (40) that performed two activities, 10 MWT and TUG. The data was recorded from eight inertial sensors. Based on the results obtained, the author demonstrated the utility of their study. The paper is well-written, presents novel results, and should be accepted for publication.
Small issues:
· Some acronyms are not defined in the text when they are first used. For example: NN. Avoid abbreviations starting within the abstract ("NN-based analysis"). All acronyms must be defined on their first appearance in the text, even if considered well-known.
· At the beginning of the paper, it is mentioned: "Forty patients underwent the 10-m walk test and the Timed Up and Go test at normal speed …" But, in the Measurement procedure subchapter the authors mentioned "Six acceleration and angular velocity signals were continuously recorded for each sensor during the TUG test. The signals include three acceleration axes …". So, what where the recording conditions for the 10-m walk test (mainly because are not presented in the text)? Is it the same as in the TUG test?
· Typo errors: "Therefore, Shorter completion" – use a small letter "s".
· In Table 2, you have a footnote reference at the True Negative (value 1) that goes nowhere. Please remove it.
· Please mention on line 247 that "4.2.2" is the version of the tool.
Author Response
Reviewer's comments are shown in black type and our responses to comments are shown in red type. Please find the detailed responses below and the corresponding corrections highlighted in red in the re-submitted files.
This is an exciting paper. The paper evaluates walking independence in acute-care hospital patients using deep neural networks (based on a transformer model). The study used a large number of subjects (40) that performed two activities, 10 MWT and TUG. The data was recorded from eight inertial sensors. Based on the results obtained, the author demonstrated the utility of their study. The paper is well-written, presents novel results, and should be accepted for publication.
Thank you for the many helpful comments. I appreciate your dedication to maintaining the standards of scholarly work.
Small issues:
- Some acronyms are not defined in the text when they are first used. For example: NN. Avoid abbreviations starting within the abstract ("NN-based analysis"). All acronyms must be defined on their first appearance in the text, even if considered well-known.
Abbreviations were removed from the abstract and corrected.
- At the beginning of the paper, it is mentioned: "Forty patients underwent the 10-m walk test and the Timed Up and Go test at normal speed …" But, in the Measurement procedure subchapter the authors mentioned "Six acceleration and angular velocity signals were continuously recorded for each sensor during the TUG test. The signals include three acceleration axes …". So, what where the recording conditions for the 10-m walk test (mainly because are not presented in the text)? Is it the same as in the TUG test?
Both tests had the same recording conditions. The indicated text has been corrected.
- Typo errors: "Therefore, Shorter completion" – use a small letter "s".
I have corrected it accordingly.
- In Table 2, you have a footnote reference at the True Negative (value 1) that goes nowhere. Please remove it.
As you pointed out, my mistake. I have deleted it.
- Please mention on line 247 that "4.2.2" is the version of the tool.
I have added it accordingly.
Reviewer 3 Report
Comments and Suggestions for Authors
The article “Evaluation of Patient's Level of Walking Independence Using Inertial Sensors and Neural Networks in an Acute-Care Hosptal” is quite interesting because working with healthy patients is difficult, but collaborating with elderly people is a challenge. In terms of walking, it is admirable because it requires much patience and, above all, dealing with people as well as being with specialists watching every step of the experiment, in addition to everything involved in the instrumentation, data collection, and processing. In general, I think it is a good job, but I have some doubts about some very particular points.
There are areas of improvement for this paper, which are listed below:
1. Could the neural network provide a way to pre-screen patients or offer objective data to supplement the physiotherapist's judgment? How?
2. A small sample size makes it challenging to know if the results would apply to a broader population.
3. The mathematical equations of gait analysis could be added. No equation of the models has been described. It is good to know which vectors are presented during the gait.
4. The authors should consider adding more explanations of the method used to program NN.
5. Figure 4 says that all are dated from the left gait cycle, but in your image, you show the right and left thing and the ankle. If you have the right gait cycle, I suggest adding this information.
6. For Table 1, what are your outputs, and what kind of information is your NN processing? Perhaps it should be referenced since I understand the architecture, but that it is processing?
7. In the conclusions from 330 -344, you state that you excluded some sensors to analyze the swing phase better and found that there is a difference between left and right during the gait analysis, so to improve that conclusion, I suggest putting an image that can illustrate that difference because I cannot find that conclusion because in your image 6, 7 , 8 and 9 that you use for your confusion matrix I cannot reach that information that you found for your conclusion.
8. I recommend you add a specific part of the conclusions.
Comments on the Quality of English LanguageNo comments
